# SGL 121 Attenuates Nonalcoholic Fatty Liver Disease through Adjusting Lipid Metabolism Through AMPK Signaling Pathway

**DOI:** 10.3390/ijms21124534

**Published:** 2020-06-25

**Authors:** Da Eun Kim, Bo Yoon Chang, Byeong Min Jeon, Jong In Baek, Sun Chang Kim, Sung Yeon Kim

**Affiliations:** 1Institute of Pharmaceutical Research and Development, College of Pharmacy, Wonkwang University, Iksan, Jeonbuk 54538, Korea; dhrtls1234@naver.com (D.E.K.); oama611@nate.com (B.Y.C.); 2Department of Biological Sciences, Korea Advanced Institute of Science and Technology, 291 Daehak-Ro, Yuseong-Gu, Daejeon 305-701, Korea; jbm0901@kaist.ac.kr (B.M.J.); baekji@kaist.ac.kr (J.I.B.); sunckim@kaist.ac.kr (S.C.K.); 3Intelligent Synthetic Biology Center, 291 Daehak-Ro, Yuseong-Gu, Daejeon 305-701, Korea; 4KAIST Institute for Biocentury, Korea Advanced Institute of Science and Technology, 291 Daehak-Ro, Yuseong-Gu, Daejeon 305-701, Korea

**Keywords:** ginsenoside F2, nonalcoholic fatty liver disease, free fatty acids, antioxidant, sterol regulatory element-binding protein

## Abstract

A ginsenoside F2-enhanced mixture (SGL 121) increases the content of ginsenoside F2 by biotransformation. In the present study, we investigated the effect of SGL 121 on nonalcoholic fatty liver disease (NAFLD) in vitro and in vivo. High-fat, high-carbohydrate-diet (HFHC)-fed mice were administered SGL 121 for 12 weeks to assess its effect on improving NAFLD. In HepG2 cells, SGL 121 acted as an antioxidant, a hepatoprotectant, and had an anti-lipogenic effect. In NAFLD mice, SGL 121 significantly improved body fat mass; levels of hepatic triglyceride (TG), hepatic malondialdehyde (MDA), serum total cholesterol (TC), high-density lipoprotein (HDL), and low-density lipoprotein (LDL); and activities of alanine aminotransferase (ALT) and aspartate aminotransferase (AST). In HepG2 cells, induced by oxidative stress, SGL 121 increased cytoprotection, inhibited reactive oxygen species (ROS) production, and increased antioxidant enzyme activity. SGL 121 activated the Nrf2/HO-1 signaling pathway and improved lipid accumulation induced by free fatty acids (FFA). Sterol regulatory element-binding protein-1 (SREBP-1) and fatty acid synthase (FAS) expression was significantly reduced in NAFLD-induced liver and HepG2 cells treated with SGL 121. Moreover, SGL 121 activated adenosine monophosphate-activated protein kinase (AMPK), which plays an important role in the regulation of lipid metabolism. The effect of SGL 121 on the improvement of NAFLD seems to be related to its antioxidant effects and activation of AMPK. In conclusion, SGL 121 can be potentially used for the treatment of NAFLD.

## 1. Introduction

In non-alcoholic fatty liver disease (NAFLD), the most common chronic liver disease, excess fat accumulates in the liver tissue [1]. In the liver, NAFLD can cause other reversible diseases, such as hepatic steatosis, steatohepatitis, and liver fibrosis, and non-reversible diseases, such as hepatocellular carcinoma and cirrhosis [2,3,4]. Treating NAFLD during the reversible phase is very important [5].

NAFLD is a serious condition that affects the development and progression of metabolic disorders, such as diabetes, obesity, and high blood pressure, making it necessary to conduct studies on NAFLD for its control and prevention [6].

The metabolic feature of note during the lesion forming process in NAFLD is the accumulation of triglycerides in liver cells [7]. Increased fat inflow into the liver due to a high-calorie or high-fat diet and increase in lipid decomposition in fat tissue due to insulin resistance cause the inflow of free fatty acids (FFA) to increase, thus leading to fat accumulation in the liver [8]. Increased blood insulin levels promote the expression of lipogenic genes in the liver [9]. FFA inflowing into the liver are converted into triglycerides through an esterification process, and then stored in the liver or contained in very-low-density lipoprotein (VLDL) and moved into the blood [10]. The accumulated lipids are known to undergo lipid peroxidation by reactive oxygen species (ROS), thereby resulting in metabolic liver disease [11]. Hepatic steatosis occurs if the inflow of fatty acids into the liver or the synthesis of fatty acids in the liver increases, or the transfer or oxidation process to the blood is inhibited [12]. Thus, controlling antioxidant activity and lipid metabolism is believed to play an important role in improving NAFLD [13,14]. Liposynthesis and metabolism are regulated by the sterol regulatory element-binding protein-1 (SREBP-1), a transcription factor [15]. SREBP-1 controls the revelation of fatty acid synthase (FAS) and acetyl CoA carboxylase (ACC), which are geosynthetic enzyme genes [16]. Moreover, the adenosine monophosphate-activated protein kinase (AMPK) controls SREBP-1 [17]. Oxidative stress and fat accumulation inhibitory effects through activation of the Nrf2/HO-1 signaling pathway have been reported [18]. Therefore, it was suggested that the Nrf2/HO-1 signaling pathway is related to NAFLD [19]. To treat patients with NAFLD, anti-obesity, insulin resistance, and hyperlipidemia drugs are used to improve risk factors; in addition, liver protectors and antioxidants are used to treat damaged liver cells [20,21]. These treatments do not work specifically on the liver, and their effects are minimal compared to the side effects that accompany them [22,23]. The development of bio-active food material and medication, which can prevent and heal NAFLD and has few side effects, is required [24,25].

Ginsenoside is the biochemically and pharmacologically active ingredient in ginseng [26]. Pharmacologically, ginsenoside acts like an anti-diabetic and anti-hypertensive, promotes liver function, cures hangovers, has an anti-fatigue, antioxidant, and anti-inflammatory activity, promotes protein synthesis, and has an immuno-enhancing effect [27,28].

Minor ginsenoside, in low-molecular-weight form (622.9–785 g/mol), is known for its higher absorption rate and high physiological activity compared to major ginsenoside (801–1109.3 g/mol) [29]; however, it is either absent or found in trace amounts in ginseng; therefore, producing minor ginsenoside has become a popular topic of research [30,31]. Minor ginsenoside F2 is known to have anti-cancer effects and anti-obesity effects [32,33]. Previously, we have produced a ginsenoside F2-enhanced mixture (SGL 121), which contains a high amount of ginsenoside F2, using biotransformation [34].

The present study aimed to determine the prevention and improvement effects of SGL 121 on NAFLD by identifying antioxidant activity and lipid metabolism regulation, using in vitro and in vivo models.

## 2. Results

### 2.1. Effects of SGL 121 on Body Weight and Body Fat

At week 12, the weight of the positive control, 200 mg/kg metformin (Met) group, was significantly reduced compared to the high-fat, high-carbohydrate (HFHC) group. A trend toward weight loss with SGL 121 was observed, but the difference was not significant (Figure 1A,B).

Dual-energy X-ray absorptiometry (DXA) was used to confirm the effect of SGL 121 on HFHC-induced body fat composition. Observed as a large red area, the relative distribution of fat mass was highest in the HFHC group. However, the SGL 121 group had a reduced fat mass distribution compared to the HFHC group (Figure 1C). In addition, the 200 mg/kg SGL 121 group showed a significant decrease in body fat mass than the HFHC group (Figure 1D).

### 2.2. The Effect of SGL 121 on Lipid Accumulation in NAFLD-Induced Mice

As shown in Figure 2A, Oil red O staining revealed fat accumulation in the liver of the HFHC mice. Fat accumulation decreased in a concentration-dependent manner by the administration of SGL 121. The hepatic triglyceride (TG) level was significantly reduced in the 100 and 200 mg/kg SGL 121 groups and was more than 60% reduced compared to that in the HFHC group (Figure 2B). No significant difference in hepatic total cholesterol (TC) level was seen; however, compared to the TC level in the HFHC group, a decrease of more than 20% was observed (Figure 2C). The serum TG levels decreased in the SGL 121 groups compared to that in the HFHC group, but there was no significant effect (Figure 2D). The serum TC levels were significantly reduced in the SGL 121 group compared to that in the HFHC group (Figure 2E). The levels of high-density lipoprotein (HDL) increased significantly in the 50 mg/kg SGL 121 group (Figure 2F). Elevated serum low-density lipoprotein (LDL) levels in HFHC-fed mice were concentration-dependent and significantly lowered by SGL 121 treatment (Figure 2G).

### 2.3. Effect of SGL 121 on Liver Function Improvement in NAFLD-Induced Mice

The activities of alanine aminotransferase (ALT), aspartate aminotransferase (AST), and γ-glutamyl transpeptidase (GGT) in the serum and the level of malondialdehyde (MDA) in the liver were measured to confirm the improvement in liver function by SGL 121 in NAFLD-induced mice, with reduced liver function. ALT and AST activities in the serum decreased significantly in the 200 mg/kg SGL 121 group and decreased by more than 30% compared to those in the HFHC group (Figure 3A,B). Serum GGT activity tended to decrease, but the difference was not significant (Figure 3C). MDA levels decreased by more than 40% in all SGL 121 groups compared to that in the HFHC group (Figure 3D).

### 2.4. Effects of SGL 121 on the Expression of Liver Fat Metabolism-Related Genes in NAFLD-Induced Mice

To investigate the mechanism of the SGL 121-mediated reduction in TG accumulation in the liver, mRNA and protein expression analysis of the regulators of lipogenesis was carried out. In the SGL 121-administered groups, mRNA expression of SREBP-1 and FAS significantly decreased compared to that in the HFHC group (Figure 4A,B). Western blot analysis also showed a dose-dependent downregulation of SREBP-1 and FAS expression in the SGL 121-administered group (Figure 4C).

### 2.5. Antioxidant Activities of SGL 121 in a Cell-Free System

2,2-diphenyl-1-picrylhydrazyl (DPPH) radical scavenging activity was evaluated to confirm the antioxidant effect of SGL 121. Ascorbic acid (50 µM), used as a positive control, showed a 95.1% DPPH radical scavenging activity. SGL 121 showed increased DPPH radical scavenging activity in a concentration-dependent manner and exhibited 48% and 79.2% scavenging activity at 250 and 500 µg/mL, respectively (Figure 5A).

For superoxide dismutase (SOD)-like activity, the positive control trolox (500 µg/mL) showed a scavenging activity of 56.2%. SGL 121 showed a significant difference compared to the control, and its radical scavenging activity of 15.7% was confirmed at 500 µg/mL, suggesting that SGL 121 showed a dose-dependent antioxidant activity (Figure 5B).

### 2.6. Cytotoxicity and Cytoprotective Effect of SGL 121

HepG2 cells were treated with SGL 121 for 24 h. Cell viability of 100 µg/mL SGL 121-treated cells was 90%, confirming no cytotoxicity at all treatment concentrations (Figure 6A). Experiments were not performed at concentrations above 200 µg/mL, which are considered cytotoxic in HepG2 cells. The cytoprotective effect of SGL 121 against tert-butyl hydroperoxide *(t*-BHP)-induced oxidative damage was evaluated. HepG2 cells were treated with SGL 121 (10, 30, and 100 µg/mL) and 100 µM *t*-BHP for 24 h. Exposure to *t*-BHP reduced cell viability by 70%. However, SGL 121 significantly increased cell viability in a dose-dependent manner, indicating protection from oxidative damage (Figure 6B).

### 2.7. Effect of SGL 121 on ROS Level and Antioxidant Activity

ROS generation was increased approximately 9-fold over the control by *t*-BHP treatment. ROS level decreased in a concentration-dependent manner and significantly after SGL 121 treatment (Figure 6C). To investigate whether the antioxidant activity of SGL 21 increased the activity of cellular antioxidant enzymes, we measured the activity of SOD in HepG2 cells treated with *t*-BHP. When cells were treated with *t*-BHP alone, the activity of SOD was significantly decreased compared to that in the control. SGL 21 increased the antioxidant activity of SOD. SOD activity was significantly increased when compared to *t*-BHP alone at 50 µg/mL or more (Figure 6D).

### 2.8. Effects of SGL 121 on HO-1 Expression and Nrf2 Nuclear Translocation in HepG2 Cells

To determine whether SGL 121 induces the expression of antioxidant enzymes, the effect of SGL 121 on HO-1 mRNA and protein expression in HepG2 cells was investigated. Analysis of HO-1 expression in HepG2 cells after treatment with 100 µg/mL of SGL 121 for 1, 6, and 12 h showed that mRNA and protein expression reached normal levels after 6 h of SGL 121 exposure (Figure 7A). As shown in Figure 7B, SGL 121 increased the mRNA and protein expression of HO-1 in a dose-dependent manner.

To investigate whether SGL 121 induced the nuclear translocation of Nrf2, time course analysis of nuclear Nrf2 expression was done under the effect of SGL 121. SGL 121 treatment induced the expression of nuclear Nrf2 protein in a time-dependent manner (Figure 7C). As shown in Figure 7D, the nuclear fractions of SGL 121-treated cells showed a gradual increase in Nrf2 expression, whereas the cytosolic fractions showed a decrease in Nrf2 expression. SGL 121 showed the greatest increase in Nrf2 expression at the 100 µg/mL treatment concentration and 6 h exposure.

### 2.9. Effects of SGL 121 on Lipid Accumulation and Lipogenesis in HepG2 Cells

HepG2 cells were treated with 1 mM FFA to induce hepatic steatosis, and stained with Oil red O to confirm intracellular lipid accumulation. FFA-induced HepG2 cells showed a significant increase in lipid content compared to that in the control. SGL 121 treatment significantly attenuated lipid accumulation in HepG2 cells (Figure 8A). To evaluate whether SGL 12 improved FFA-induced hepatic steatosis, expression of AMPK, PPARα, SREBP-1, FAS, and ACC—genes related to lipid metabolism—was evaluated. SGL 121 treatment significantly increased the phosphorylation of AMPK and ACC (Figure 8C). The mRNA and protein levels of PPARα, associated with β-oxidation, were significantly increased by SGL 121 treatment in FFA-induced HepG2 cells (Figure 8B,C). SGL 12 treatment was observed to significantly reduce the mRNA and protein expression levels of SREBP-1 and FAS, compared to the FFA treatment group (Figure 8C).

## 3. Discussion

The incidence of fatty liver disease is rapidly increasing due to an increase in the incidence of obesity and diabetes in the population. In particular, the proportion of NAFLD associated with obesity, diabetes, and metabolic syndromes accounts for a majority of fatty liver diseases [35,36].

Ginsenoside can be largely divided into the protopanaxadiol (PPD) and protopanaxatriol (PPT) types depending on their chemical structure. Rb1, Rb2, Rb3, Rc, Rd, Rg3, and Rh2 are PPD-type ginsenosides, whereas Rg1, Rg2, F1, and Rh1 are the PPT-type ginsenosides [37]. In addition, depending on the content of ginseng, ginsenosides can also be classified into two categories, major and minor ginsenosides. Minor ginsenoside is difficult to procure and, thus, hard to use industrially. Recently, the acquisition of minor ginsenoside has become relatively easy due to synthetic biological methods, such as the development of a conversion technique using microbial enzymes and recombinant enzymes, and recombinant ginseng-effective microbiology utilization [38,39]. SGL 121 was created through the biotransformation of PPD-type ginsenoside [34]. NAFLD-related studies have been conducted in various ginsenosides, such as compound K, ginsenoside Rh1, and Rb1 [40,41]. SGL 121, used in this study, contains a high amount of ginsenoside F2. Ginsenoside F2 and compound K have several pharmacological properties [42,43,44]. Thus, using in vitro and in vivo models, we have discovered that SGL 121 could cure NAFLD. NAFLD was induced in the mice using a high-fat, high-carbohydrate diet. We observed the effect of SGL 121 administration on the pathological process of NAFLD. C57BL/6 mice were divided into the HFHC group, SGL 121 groups, and the metformin group, used as a positive control, and administered SGL 121 or metformin for 12 weeks to investigate their effect on lipid levels, liver function, degree of fatty liver, and factors related to liver lipogenesis. SGL 121 did not cause a significant weight loss, but fat distribution and body fat weight were significantly decreased, as seen by the DXA-body composition analysis (Figure 1). Histopathological observations suggested that SGL 121 decreased liver fat in a concentration-dependent manner (Figure 2A). SGL 121 administration decreased hepatic TG and TC levels compared to those in the HFHC group, and hepatic TG levels were significantly different (Figure 2B,C). Serum TG and TC levels decreased compared to those in the HFHC group, and serum TC levels were significantly different (Figure 2D,E). Serum LDL levels also decreased in a concentration-dependent manner and were significantly lower than those in the HFHC group (Figure 2F). Lipid-lowering therapy such as statin, fibrate, and ezetimibe is reported to improve the liver biochemistry and histology in patients with NAFLD [45,46,47]. SGL 121 showed a significant anti-lipogenic effect on NAFLD of the in vivo model. It is expected to have a positive effect on lipid-lowering therapy in patients with NAFLD. ALT and AST activities in the serum and MDA levels in the liver were significantly decreased compared to those in the HFHC group. Serum GGT activity tended to decrease (Figure 3). Western blot and real-time PCR were performed to confirm the effect of SGL 121 on regulators of fat synthesis. SGL 121 decreased the expression of SREBP-1 and FAS in a concentration-dependent manner, and this difference was significant (Figure 4). Ginsenoside lowers cholesterol levels by promoting the absorption and metabolism of fats and cholesterol and by reducing the synthesis inhibition of LDL receptors in high-fat-diet models [48]. According to previous reports, ginseng and ginsenoside Rb1, Rb2, Re, compound K, Rg1, Rg2, Rk3, and Rg3 alleviate liver lipid accumulation through AMPK activation [49,50]. Activation of AMPK regulates the expression of SREBP-1, FAS, ACC, and stearoyl CoA desaturase-1, which is involved in lipogenesis, and the expression of SREBP-2 and HMG-CoA reductase involved in cholesterol synthesis [51]. Similar to ginsenoside in improving lipid metabolism and lipogenesis, SGL 121 alleviates fatty liver disease by improving lipid metabolism and inhibiting the lipid synthesis pathway through SREBP-1. SGL 121 is expected to relieve liver lipid accumulation associated with AMPK activation.

The liver, one of the main internal organs of the body, is responsible for detoxification and plays an important role in metabolic pathways, such as protein synthesis and nutrient storage [52]. The liver is inflamed and damaged by oxidative stress, such as ROS. Furthermore, oxidative stress causes metabolic diseases, cancer, and aging. Tests conducted in high-fat-diet-fed obese animals have shown that the imbalance between antioxidant defense systems and ROS causes tissue damage [53].

Normally, ROS produced in vivo is eliminated by ROS-eliminating enzymes, such as superoxide dismutase (SOD), catalase, glutathione peroxidase, and glutathione S-transferase (GST), or by antioxidant enzymes such as reduced glutathione (GSH), ascorbic acid, and tocopherol [54].

Here, the antioxidant activity and liver protection effects of SGL 121 have been evaluated. The antioxidant effect of SGL 121 showed a concentration-dependent increase for all methods (Figure 5 and Figure 6D). In HepG2 cells, in which oxidative stress was induced with *t*-BHP, SGL 121 showed concentration-dependent protection of the tissue (Figure 6B) and concentration-dependent suppression of ROS production (Figure 6C).

Nuclear factor erythroid 2-related factor-2 (Nrf2) is a transcription factor that protects cells from oxidative stress. Under basal conditions, it forms a complex with Keap1 in the cytoplasm, and is activated by stimulation due to antioxidants or activators. Nrf2 separates from the complex and enters the nucleus to promote the synthesis of antioxidant enzymes, such as GST, SOD, catalase, and heme oxygenase-1 (HO-1), which can protect from ROS generation and cytotoxicity [55,56]. In the experimental model of NAFLD, Nrf2 is involved in liver fat metabolism-related genes and liver fatty acid metabolism; Nrf2 activation has been reported to reduce oxidative stress and prevent hepatic steatosis [56]. Therefore, to measure the effect of SGL 121 on the entry of Nrf2 into the nucleus, the expression of Nrf2 was measured by separating the cell extract into a nuclear fraction and a cytoplasmic fraction. SGL 121 increased the translocation of Nrf2 into the nucleus. These results showed that SGL 121 increases the expression of HO-1 by promoting the entry of Nrf2 into the nucleus (Figure 7). It has been reported that HO-1 expression in hepatocytes not only provides resistance to oxidative stress but also effectively eliminates inflammation of the liver, thereby protecting the liver [57]. Activation of HO-1 and SOD enzymes by SGL 121 is thought to contribute significantly to the inhibition of fat accumulation by regulating ROS production.

In a model in which HepG2 cells were treated with FFA to induce hepatic steatosis, the effect of SGL 121 on lipid content and expression of fat metabolism-related factors was investigated. The increase in lipid content and SREBP-1 and FAS expression due to FFA-induction showed a concentration-dependent and significant inhibitory effect by SGL 121 (Figure 8). These results indicated that SGL 121 downregulates fat production markers in FFA-induced HepG2 cells, and suggests that SGL 121 suppresses TG accumulation by reducing lipid accumulation. Peroxisome proliferator-activated receptor α (PPARα) regulates fatty acid oxidation and gene expression related to lipid metabolism [58]. PPARα activation increases fatty acid transport into the mitochondria and increases the expression of acyl-CoA oxidase 1 and CPT-1 to increase fatty acid oxidation [59]. Reduction of PPARα expression in the liver of patients with fatty livers has been reported; PPARα agonists, such as resveratrol, naringenin, *trans*-caryophyllene, and quercetin, are known to improve fatty liver disease and nonalcoholic fatty liver disease [60,61]. PPARα gene expression level was significantly increased after SGL 121 treatment (Figure 8B,C). SGL 121 is thought to improve lipid accumulation in hepatocytes by inducing PPARα activation. Phosphorylation of AMPK and ACC was significantly increased upon SGL 121 treatment. These results suggest that SGL 121 may act to reduce hepatic lipidosis by activating AMPK.

Assays for evaluating the DPPH radical scavenging activity, SOD-like activity, antioxidant enzyme activity, Nrf2/HO-1 pathway activity, and liver protection effect in in vivo and in vitro models proved that SGL 121 has excellent antioxidant activity. Moreover, SGL 121 can prevent or improve NAFLD by reducing body fat mass, improving lipid levels in the liver and serum, and reducing mRNA and protein expression levels of lipogenic factors.

Taken together, the results showed that SGL 121 had a significant hepatoprotection anti-lipogenic effect in the NAFLD model, and reduced lipogenesis in the liver by inhibiting the activity of antioxidants and the expression of factors that induce lipogenesis. SGL 121 suggests the potential to improve lipids and reduce liver lipogenesis in NAFLD. However, in vivo and in vitro models have many limitations [62]. SGL 121 is currently undergoing clinical trials for NAFLD. In addition, we will research metabolic syndrome and liver fibrosis.

Through further research, SGL 121 can be developed as a health food for the prevention or improvement of NAFLD and metabolic diseases of the liver.

## 4. Materials and Methods

### 4.1. Materials

Fetal bovine serum (FBS), DMEM, penicillin/streptomycin (PC/SM), and phosphate-buffered saline (PBS) were obtained from Invitrogen Inc. (Carlsbad, CA, USA). Sodium palmitate, sodium oleate, pioglitazone hydrochloride, Oil red O solution, metformin, 3-(4,5-dimethylthiazol-2-yl)-2,5-diphenyltetrazolium bromide (MTT), tert-butyl hydroperoxide (*t*-BHP), bovine serum albumin, and 2,2-diphenyl-1-picrylhydrazyl (DPPH) were purchased from Sigma-Aldrich (St. Louis, MO, USA). The SOD Assay Kit-WST was purchased from Dojindo Laboratories (Kumamoto, Japan). Radioimmunoprecipitation assay (RIPA) buffer, TaqMan RNA-to-Ct™ 1-step kit, and TaqMan^®^ Gene Expression Assays were obtained from Thermo Fisher Scientific (Grand Island, NY, USA). Antibodies were purchased from Enzo Life Sciences (Farmingdale, NY, USA), Santa Cruz Biotechnology Inc. (Santa Cruz, CA, USA), and Cell Signaling Technology (Beverly, MA, USA).

### 4.2. Preparation of SGL 121

SGL 121 was prepared from ginsenosides, using the biotransformation process. The process used immobilized whole cells designed to over-express β-glucosidase (BglPm) in *Corynebacterium glutamicum* ATCC 13032 [34].

### 4.3. Animal Experiment

Five-week-old C57BL/6 male mice were obtained from Orientbio Korea Inc. (Sungnam, Korea) The animals were housed under optimal conditions of humidity (50–55%) and temperature (22–25 °C) with a 12 h light/dark cycle and were given free access to food and water.

The experimental diet used to induce NAFLD was a high-fat, high-carbohydrate (HFHC) diet (60% fat and 42 g carbohydrates/L drinking water, ratios at 45% sucrose and 55% fructose) [63]. The HFHC diet was provided for 12 weeks and body weight and food intake were measured once a week.

The mice were divided into the HFHC group (HFHC), HFHC + SGL 121 (50, 100, and 200 mg/kg) group, and the HFHC + metformin (200 mg/kg, positive control) group.

SGL 121 and metformin were administered orally five times a week for 12 weeks. Blood was collected from the portal vein. Liver tissues were stored at –80 °C. All animal procedures were approved by Wonkwang University, following guidelines for the care and use of laboratory animals (approved number: WKU19-02, approved date: 12 February 2019).

### 4.4. Biochemical Assays

Serum triglycerides (TG, Biovision, Milpitas, CA, USA), total cholesterol (TC, Biovision, Milpitas, CA, USA), high-density lipoprotein cholesterol (HDL-C, Biovision, Milpitas, CA, USA), γ-glutamyl transpeptidase (GGT, Biovision, Milpitas, CA, USA), aspartate aminotransferase (AST, Asan Pharm Co. Ltd., Seoul, Korea), and alanine aminotransferase (ALT, Asan Pharm Co. Ltd., Seoul, Korea) levels were measured using commercial kits. Quantification was carried out according to the manufacturer’s protocol. The level of LDL-C was calculated using the Friedewald equation [64].

### 4.5. Hepatic Biochemical Analysis

A total of 0.1 g of liver tissue was homogenized in an ice-cold homogenizing buffer that contained 1.15% KCl, 50 mM Tris, and 1 mM EDTA (pH 7.4). Hepatic triglyceride (TG, Biovision, Milpitas, CA, USA), total cholesterol (TC, Biovision, Milpitas, CA, USA), and lipid peroxidation (MDA, Biovision, Milpitas, CA, USA) levels were determined using commercial kits. Quantification was carried out according to the manufacturer’s protocol.

### 4.6. Body Fat Composition Analysis

A body fat scan was performed using dual-energy X-ray absorptiometry (DXA) with a complete body scanner (InAlyzer DXA, Medikors, Seongnam, Korea), as per the manufacturer’s instructions.

### 4.7. Histopathology

The liver was fixed in 4% formalin followed by 30% sucrose for histological analysis. After the tissue was cryo-embedded, 10 µm thickness cryosections were cut at −20 °C. The sections were stained with Oil red O/hematoxylin and visualized under a microscope (Leica Microsystems, Wetzlar, Germany).

### 4.8. DPPH Free Radical Scavenging Activity

A solution of 0.1 mM DPPH was prepared in methanol and mixed with various concentrations of SGL-121 in a 1:1 ratio. The reaction was carried out in the dark for 30 min and absorbance was measured at 520 nm. Ascorbic acid was used as a positive control.

### 4.9. Superoxide Radical Scavenging Activity

SGL-121, diluted at various concentrations, or protein samples obtained after treatment with *t*-BHP in cells were used for the experiment. Measurement of SOD-like activity was carried out using a commercially available SOD assay kit (Dojindo Molecular Technologies, Kumamoto, Japan).

### 4.10. Cell Culture and Cytotoxicity Assay

HepG2 cells were obtained from the American Type Culture Collection (ATCC, Manassas, VA, USA) and maintained in DMEM, 1% PC/SM, and 10% FBS in 5% CO_2_ at 37 °C. Cytotoxicity of *t*-BHP, SGL-121, or both was measured using the MTT assay, where *t*-BHP induces oxidative stress. Cells were seeded at 5 × 10^4^ cells/well in a 96-well plate and treated with *t*-BHP in the presence or absence of SGL 121. After a 24 h incubation period, 1 mg/mL MTT solution was added and the plate was incubated for 3 h. The absorbance was measured at 540 nm. Silymarin was used as a positive control.

### 4.11. Measurement of ROS Level

HepG2 cells were treated with SGL-121 for 24 h and oxidative stress was induced with 100 µM *t*-BHP for 2 h. The level of intracellular ROS was quantified using the oxidation-sensitive fluorescent probe, DCFH-DA. Fluorescence intensities were measured at an excitation wavelength of 490 nm and an emission wavelength of 525 nm (BioTek, Winooski, VT, USA). Silymarin was used as a positive control.

### 4.12. Measurement of Lipid Content with Oil Red O Staining

HepG2 cells were treated with FFA (oleate:palmitate, 2:1 ratio) in the presence or absence of SGL-121. After a 24 h incubation period, the cells were washed twice with PBS and fixed with 4% paraformaldehyde for 1 h at room temperature. The cells were then washed three times with PBS and stained with Oil red O solution for 20 min. The plate was washed and the lipid-stained red dye was dissolved using 100% isopropanol. Absorbance was measured at 510 nm. Results were expressed as the percentage of Oil red O-stained material compared to the control.

### 4.13. Analysis of mRNA Expression

Total RNA was extracted from the HepG2 cells and homogenized liver tissue. A 20 µL PCR reaction mixture, containing RNA, specific probes, reverse transcriptase, and DNA-polymerase, was used to perform PCR, according to the manufacturer’s recommendations (TaqMan RNA-to-CT 1 step kit). Gene expression was determined by the 2^–ΔΔCT^ method using StepOne Software (Applied Biosystems, Foster City, CA, USA).

### 4.14. Western Blotting

HepG2 cells and liver tissue were lysed in RIPA buffer, containing the protease inhibitor cocktail, for 30 min. The supernatant was obtained after centrifuging the lysate for 25 min at 13,000 rpm and 4 °C. Proteins (20–50 µg) were subjected to SDS polyacrylamide gel electrophoresis and were transferred to membranes. The membranes were blocked for 1 h and then probed with primary antibodies. After the membranes had been washed, they were incubated with secondary antibodies for 1.5 h. The proteins were detected using an ECL detection kit and visualized using the FluorChem E system image analyzer (Cell Biosciences, Santa Clara, CA, USA). β-actin was used as an internal control. The intensities of the protein bands were measured using Image J software (NIH, Bethesda, MD, USA).

### 4.15. Statistical Analysis

Statistical calculations were performed using the SigmaPlot software, version 10.0 (Systat Software Inc., San Jose, CA, USA). The results are expressed as the mean ± SE. Student’s *t*-test was used for statistical analyses; *p*-values <0.05 were considered statistically significant.

## Figures and Tables

**Figure 1 ijms-21-04534-f001:**
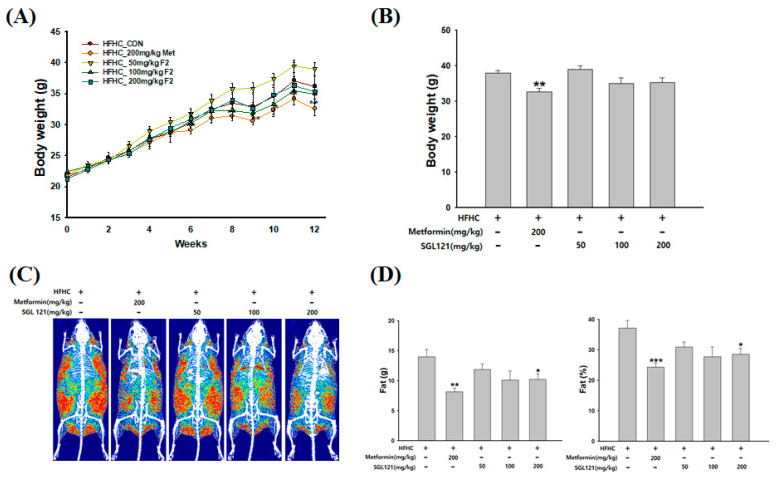
Effects of SGL 121(ginsenoside F2-enhanced mixture) on body weight and fat mass in nonalcoholic fatty liver disease (NAFLD)-induced mice. (**A**) Bodyweight changes, (**B**) bodyweight at week 12, (**C**) dual-energy X-ray absorptiometry (DXA) image of fat distribution, and (**D**) fat mass. Data are presented as the mean ± SE, n = 8. * *p* < 0.05, ** *p* < 0.01, and *** *p* < 0.001 compared to the high-fat, high-carbohydrate (HFHC) group.

**Figure 2 ijms-21-04534-f002:**
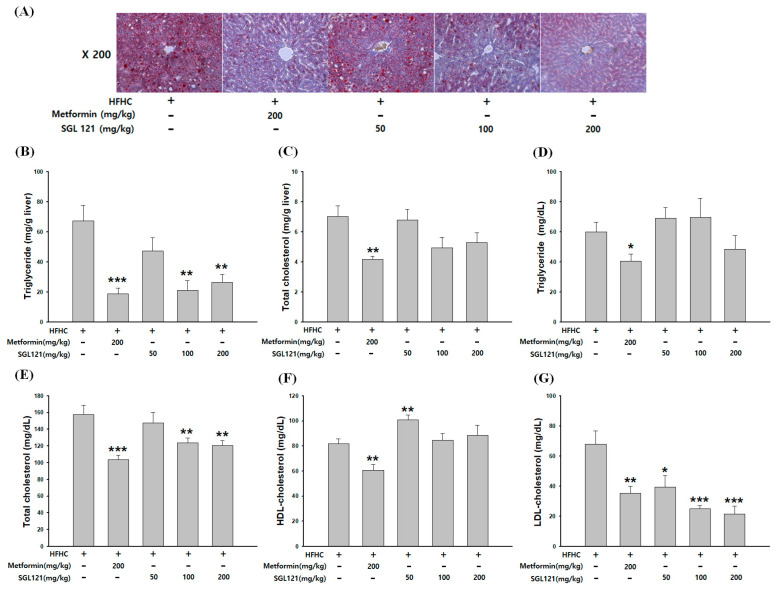
Effect of SGL 121 on lipid accumulation in NAFLD-induced mice. (**A**) Oil red O or hematoxylin staining of the liver, (**B**) hepatic triglyceride (TG), (**C**) hepatic total cholesterol (TC), (**D**) serum TG, (**E**) serum TC, (**F**) high-density lipoprotein (HDL) cholesterol, and (**G**) low-density lipoprotein (LDL) cholesterol levels. Data are presented as the mean ± SE, n = 8. * *p* < 0.05, ** *p* < 0.01, and *** *p* < 0.001 compared to the HFHC group.

**Figure 3 ijms-21-04534-f003:**
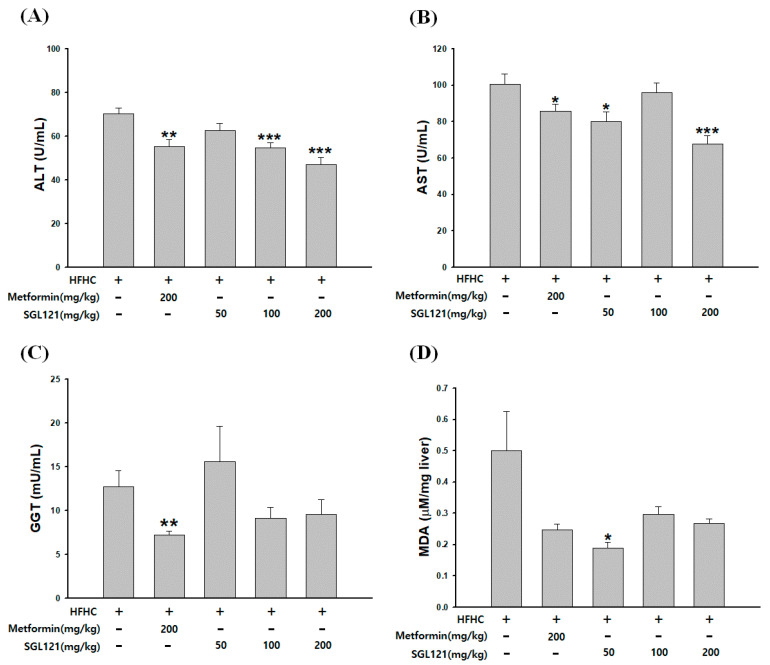
Effect of SGL 121 on liver function improvement in NAFLD-induced mice. (**A**) Serum aspartate aminotransferase (AST) activity, (**B**) serum AST activity, (**C**) serum γ-glutamyl transpeptidase (GGT) activity, and (**D**) malondialdehyde (MDA) level in liver. Data are presented as the mean ± SE, n = 8. * *p* < 0.05, ** *p* < 0.01, and *** *p* < 0.001 compared to the HFHC group.

**Figure 4 ijms-21-04534-f004:**
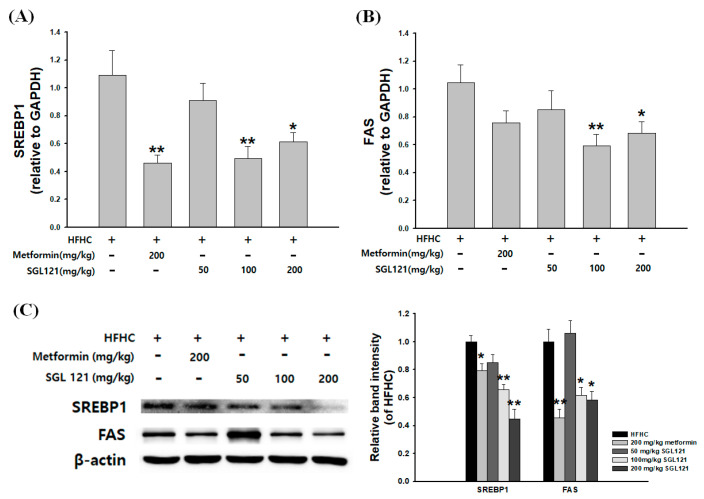
Effects of SGL 121 on hepatic lipid gene expression in NAFLD-induced mice. (**A**) sterol regulatory element-binding protein-1 (SREBP-1) mRNA levels, (**B**) fatty acid synthase (FAS) mRNA levels, (**C**) SREBP-1 and FAS protein expression levels. The graph indicates the expression level against β-actin expression level. Data are presented as the mean ± SE, n = 8. * *p* < 0.05 and ** *p* < 0.01 compared to the HFHC group.

**Figure 5 ijms-21-04534-f005:**
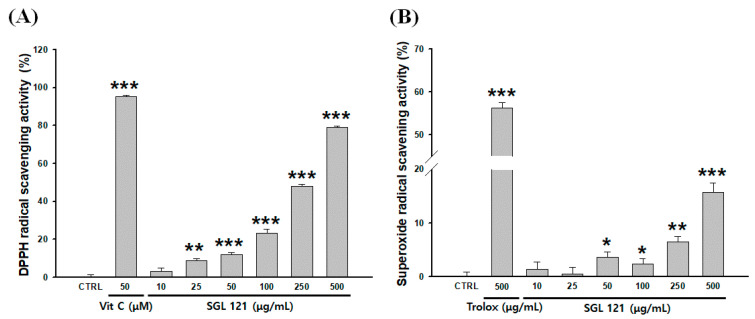
Antioxidant activities of SGL 121 on various radicals determined by (**A**) 2,2-diphenyl-1-picrylhydrazyl (DPPH) radical scavenging activity and (**B**) superoxide dismutase (SOD) radical scavenging activity. Vitamin C (A) and trolox (B) were used as positive controls. Data are presented as the mean ± SD. * *p* < 0.05, ** *p* < 0.01, and *** *p* < 0.001 compared to the control.

**Figure 6 ijms-21-04534-f006:**
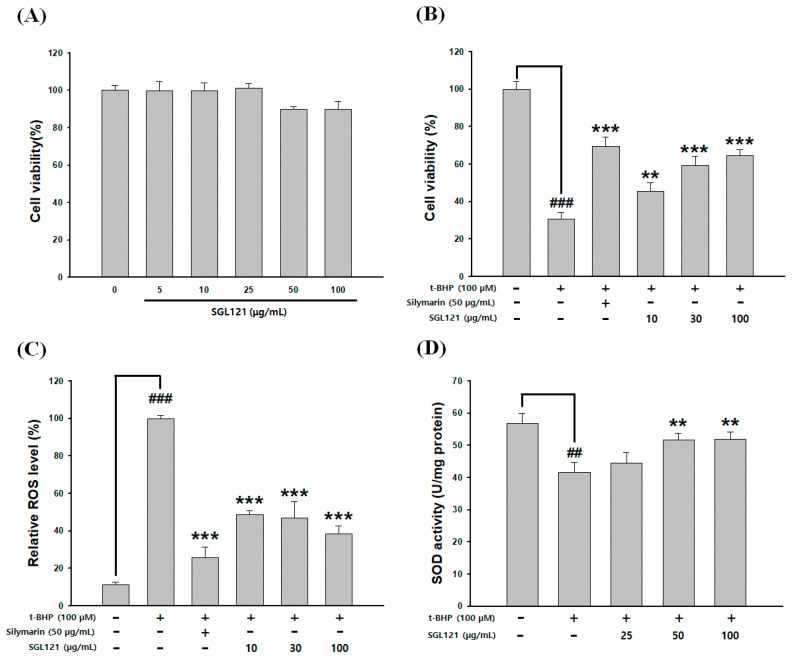
Effects of SGL 121 on tert-butyl hydroperoxide *(t*-BHP)-induced (**A**) cytotoxicity, (**B**) hepatotoxicity, (**C**) inhibition of reactive oxygen species (ROS) production, and (**D**) antioxidant enzymes. Data are presented as the mean ± SD. ## *p* < 0.01 and ### *p* < 0.001 compared to the control. ** *p* < 0.01 and *** *p* < 0.001 compared to *t*-BHP treatment alone.

**Figure 7 ijms-21-04534-f007:**
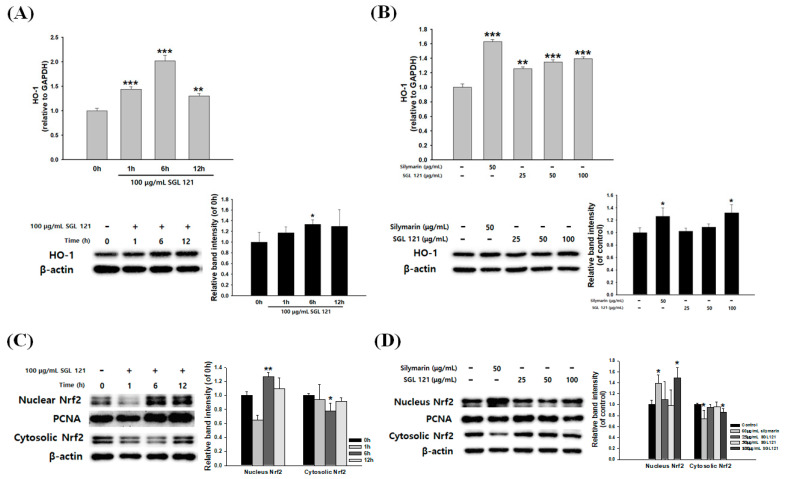
Effects of SGL 121 on HO-1 expression and Nrf2 nuclear translocation in HepG2 Cells. (**A**) Time-course analysis of mRNA and protein expression of HO-1 at 100 µg/mL SGL 121, (**B**) analysis of mRNA and protein expression of HO-1 after 6 h treatment of 25, 50, and 100 µg/mL SGL 121, (**C**) time course analysis of Nrf2 levels in nuclear and cytosolic extracts at 100 µg/mL SGL 121, and (**D**) analysis of Nrf2 levels in nuclear and cytosolic extracts after 25, 50, and 100 µg/mL SGL 121 treatment for 6 h. Data are presented as the mean ± SD. * *p* < 0.05, ** *p* < 0.01 and *** *p* < 0.001 compared to the control.

**Figure 8 ijms-21-04534-f008:**
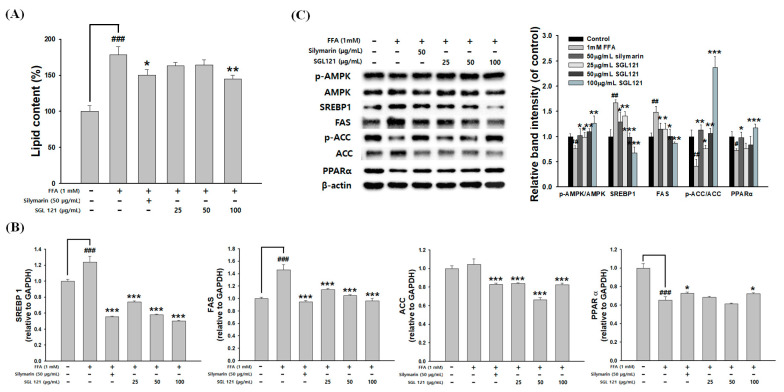
Effects of SGL 121 on lipid accumulation and lipogenesis in HepG2 Cells. (**A**) Lipid accumulation of Oil red O contents was quantified by spectrophotometric analysis, (**B**) relative mRNA levels of SREBP-1, FAS, acetyl CoA carboxylase (ACC), and peroxisome proliferator-activated receptor α (PPARα) were assayed by real-time PCR analysis, and (**C**) protein expression of p-adenosine monophosphate-activated protein kinase (AMPK), AMPK, SREBP-1, FAS, p-ACC, ACC, and PPARα was determined by Western blotting. The graph indicates the expression level against β-actin expression level. Data are presented as the mean ± SD. ## *p* < 0.01 and ### *p* < 0.001 compared to the control. * *p* < 0.05, ** *p* < 0.01 and *** *p* < 0.001 compared to free fatty acids (FFA) treatment alone.

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
