# Peer review of "SGL 121 Attenuates Nonalcoholic Fatty Liver Disease through Adjusting Lipid Metabolism Through AMPK Signaling Pathway"

_ijms, 2020, doi:10.3390/ijms21124534_

Round 1

Reviewer 1 Report

In this study the authors showed the hepatoprotective role of ginsenoside (SGL 121) by reducing NAFLD in a mouse model. The experimental study is interesting and well described; however, it may be modified to better described the obtained results.

In the Introduction, the authors don't focused on the key role of insuline resistance and related lipid abnormalities in NAFLD; please consider related articles and comment these in the Introduction.

In the Discussion:

- The authors showed the anti-lipogenic effect of the SGL121 such as statins and fibrate. However, tha authors may comment the beneficial effect of lipid lowering therapya in patients with NAFLD; please consider related studies and comment these in the Discussion. 

- The authors may mentioned the limitations of the study; the authors may comment the limitation of the type of the study (in vitro and in vivo mouse model) and it may be interesting to evaluate the potential of these new compounds in human studies. 

Author Response

Dear Reviewer

It is with excitement that we were granted the opportunity to resubmit to you a revised version of manuscript Title: ‘SGL 121 Attenuates Nonalcoholic Fatty Liver Disease through Adjusting Lipid Metabolism through AMPK Signaling Pathway’. We have highlighted the changes in the text for the ease of your review.

Best regards,

Reviewer 2 Report

Kim et al. present the novel compound SGL 121 that may be useful in the treatment of nonalcoholic fatty liver disease. They compare the effects of SGL 121 to metformin, which is the current state of the art treatment, in an in vivo model of NAFLD. For in vitro analyses Kim et al. used the hepatoma cell line HepG2. However, the experimental setup has some drawbacks.

The authors chose to start treatment together with high fat diet. The conclusion that SGL 121 improves NAFLD is therefore inaccurate due to the study design, which has more a prophylactic character. To support their conclusions, authors could start treatment with SGL 121 in a model with established NAFLD.

The authors used HepG2 cells for their in vitro investigations, which is a liver cancer cell line. Therefore, the authors should repeat some key experiments with primary hepatocytes to strengthen their findings.

The authors describe many features of NAFLD and effects of ginsenosides in the introduction but did not test them in their study. Therefore it would be improve the study if the authors could present some markers for metabolic disorders, inflammation and maybe also fibrosis and the effect of SGL 121 on these.

The joint presentation of qPCR and representative Western blots is often misleading, especially since they seem not to show the same direction in some cases. Therefore, quantification of protein expression should be presented as well and discrepancies between gene and protein expression should be discussed.

In Figure 5, the SGL 121 doses of 250µg/mL and 500µg/mL show the strongest anti-oxidant effect. However, both doses are not presented in the following experiments. Therefore, they should be removed from the figure or authors should discuss why they not used the doses with the strongest effects in the following experiments. Although the anti-oxidant effects of the lower doses are significant in to untreated controls, their effect is only minor.

The Nrf2/HO-1 pathway should be explained in the introduction.

The first line of the discussion seems to be a text block from the manuscript template.

Author Response

(The authors gave the same response as above.)

Reviewer 3 Report

Major comments:

  • In fig 1D, is the fat mass normalized to body weight?
  • What are the levels of MDA in serum/plasma? Are they significant?
  • Please quantify the western blots.
  • Mention in brief what is t-BHP and Silymarin are in your Methods.
  • Fig 6A: It is a bit interesting that the cell viability goes down to 90% at higher concentrations of SGL121, please explain

Minor comments:

  • Line 47: ‘..are known to undergo lipid peroxidation…’
  • Lines 61-70: Mention the molecular weights of ginsenoside and minor ginsenoside.
  • Increase the font of the text in figures
  • Line 96: ‘The levels of HDL increased significantly….’
  • Need to fix Fig 6B. Labeling is misplaced.

Author Response

Dear Reviewer

It is with excitement that we were granted the opportunity to resubmit to you a revised version of manuscript Title: ‘SGL 121 Attenuates Nonalcoholic Fatty Liver Disease through Adjusting Lipid Metabolism through AMPK Signaling Pathway’. We have highlighted the changes in the text for the ease of your review.

Best regards

Round 2

Reviewer 2 Report

no further comments